# Comparison of the Complications, Reoperations, and Clinical Outcomes between Open Reduction and Internal Fixation and Total Elbow Arthroplasty for Distal Humeral Fractures in the Elderly: A Systematic Review and Meta-Analysis

**DOI:** 10.3390/jcm11195775

**Published:** 2022-09-29

**Authors:** Hyun-Gyu Seok, Jeong-Jin Park, Sam-Guk Park

**Affiliations:** Department of Orthopedics, Yeungnam University Medical Center, Daegu 41000, Korea

**Keywords:** total elbow arthroplasty, total elbow replacement, distal humeral fracture, elderly, open reduction and internal fixation, plate osteosynthesis, meta-analysis

## Abstract

Distal humeral fractures are challenging injuries seen in the elderly. Open reduction and internal fixation (ORIF) are the gold standard treatments. Total elbow arthroplasty (TEA) is an alternative to ORIF. This study aimed to pool and analyze the outcomes and complications in elderly patients with distal humeral fractures treated with either ORIF or TEA by performing a meta-analysis. We searched the PubMed, Embase, Google Scholar, and Cochrane Library databases for studies that compared the clinical and functional outcomes of ORIF and TEA in patients aged 60 years or older. After screening and performing a quality assessment of the articles, we obtained one randomized control study and nine retrospective comparative studies. The odds ratio and standardized mean difference were used to analyze the differences in outcomes between the two surgical options. In terms of the flexion/extension arc, TEA produced significantly better outcomes than ORIF (*p* = 0.02). The rates of reoperation and elbow stiffness were significantly lower in the TEA group than in the ORIF group (*p* = 0.003 and *p* = 0.04, respectively). However, the functional scores and other ranges of motion (flexion, loss of extension, pronation, supination) after surgery were similar between the two groups. The outcomes from the present meta-analysis can provide guidance when selecting a surgical option for distal humeral fractures in the elderly.

## 1. Introduction

Distal humeral fractures represent a relatively small proportion of adult fractures (~1–2%) [1,2]. This injury has a bimodal distribution with a peak incidence in young males, secondary to high-energy trauma, and the second peak in osteoporotic elderly patients [1,3,4]. Because the population is aging, it is predicted that the incidence of distal humeral fractures in the elderly population will increase [1,3]. Open reduction and internal fixation (ORIF) with double-locking osteosynthesis have become the gold standard of treatment for intra-articular distal humeral fractures [5,6,7,8]. It is difficult for elderly patients to obtain stable fixation and satisfactory functional outcomes with ORIF, owing to poor bone quality, comorbidities, and poor compliance [9,10,11]. An alternative surgical option for distal humeral fractures is total elbow arthroplasty (TEA) [12]. Initially, TEA showed good results when used to treat rheumatoid arthritis [13,14]. Currently, with the development of prostheses and techniques, TEA is a well-accepted option for the surgical treatment of other pathological conditions of the elbow. Also, promising results have been achieved following the use of TEA for complex fractures of the distal humerus, and the indications for this procedure are growing [14]. TEA is associated with complications, such as infection, dislocation, aseptic loosening, and periprosthetic fractures [15,16,17]. Prior meta-analyses have compared the outcomes of these surgical options. Due to a lack of comparative studies, these analyses had limitations in that an accurate comparison was difficult because the analysis conducted included studies that dealt only with either TEA or ORIF [1,18,19]. Recently, comparative studies comparing the two techniques have been published [5,20,21]. This study aimed to provide an updated meta-analysis, examining the outcomes by performing a meta-analysis using only comparative studies.

## 2. Materials and Methods

This meta-analysis was conducted in accordance with Preferred Reporting Items for Systematic Reviews and Meta-Analysis (PRISMA) guidelines [22].

### 2.1. Literature Search Strategy and Study Selection

We systematically searched for relevant articles in PubMed, Embase, Google Scholar, and Cochrane Library on studies published between 1 January 2000 to 30 June 2022. The following search terms were used in the research: (“distal humeral fracture” or “intercondylar humeral fracture”), (“total elbow arthroplasty” or “total elbow replacement”), and (“open reduction and internal fixation” or “plate osteosynthesis”).

We applied the following inclusion criteria for the selection of articles: (1) preoperative condition: distal humeral fracture requiring surgery in the elderly (older than 60 years old); (2) surgical method: ORIF and TEA; (3) quantitative studies, such as comparative and randomized controlled studies; (4) studies with adequate data for analysis; and (5) studies published in English.

The exclusion criteria were as follows: (1) case reports, reviews, or other indistinct forms; (2) studies that repeatedly published the same data; and (3) studies with no reports on study outcomes.

### 2.2. Data Extraction

After discarding duplicate studies, two reviewers (S.G.P. and H.G.S.) independently evaluated the potentially eligible studies. The remaining studies were screened for eligibility based on a review of the titles and abstracts. After screening, the eligible articles were independently read in full by the two authors, and the eligibility of each article was reassessed. Disagreements were resolved through consensus. Conflicts were resolved by including a third author (J.J.P.). Subsequently, data including the first author, publication year, study design, demographic information, number of patients, follow-up duration, surgical technique, and outcomes (functional scores, range of motion, and complications), were extracted.

### 2.3. Quality Assessment

The methodological quality of the included studies was assessed using the Newcastle–Ottawa Scale (NOS). The NOS evaluates the quality of studies via three aspects: selection of subjects, comparability of groups, and assessment of the outcomes [23]. The quality of each study was graded as good, fair, or poor. All studies evaluated by NOS were confirmed to be of good quality (Table 1).

### 2.4. Statistical Analyses

RevMan 5.4 software (Cochrane, London, UK) was used for the statistical analyses of the pooled data. To measure the extent of inconsistency among the results, a heterogeneity test was performed during each analysis, using I^2^ statistics. An I^2^ value of <50% indicated homogeneity of the pooled data. The fixed-effects model was used for the analysis. In contrast, when the I^2^ value was ≥50%, the pooled data were considered heterogeneous, and a random-effects model was applied.

We analyzed the odds ratio (OR) to identify the differences in reoperation and complication rates between the ORIF and TEA groups. In addition, we used the standardized mean difference (SMD) to analyze continuous data, such as the Mayo Elbow Performance score (MEPS), Disabilities of the Arm, Shoulder, and Hand (DASH), and range of motion (functional arc, flexion, loss of extension, pronation, and supination). Additionally, 95% confidence intervals (CIs) were used in the analysis. A *p*-value of <0.05 was considered significant.

## 3. Results

### 3.1. Study Selection and Characteristics

Figure 1 shows a flowchart of the screening and detailed selection process. Of the initial 300 articles, 128 were duplicates and were excluded. The titles and abstracts of the remaining articles were reviewed for the initial screening, and 21 articles were considered appropriate for the next stage of review. After a detailed assessment, 11 articles were excluded based on the inclusion and exclusion criteria. Finally, 10 studies (nine were retrospective comparative studies, and one was a randomized controlled study) were included in our meta-analysis. The selected 10 studies included 977 cases for the ORIF group and 358 cases for the TEA group. The detailed characteristics of each study are listed in Table 1.

### 3.2. Meta-Analysis Results

#### 3.2.1. Functional Scores

Postoperative elbow scores were reported in seven studies. The MEPS and DASH are the most commonly used. Figure 2 shows the forest plots, SMD, 95% CI, and heterogeneity of the functional scores. Four studies [5,10,24,27] compared TEA and ORIF using MEPS scores, and four studies [5,10,21,27] compared two groups using DASH; the random-effect model was used for the analysis of the clinical outcomes (MEPS, I^2^ = 89% and DASH, I^2^ = 92%). As a result of the analysis, there was no statistically significant difference in the functional scores between the two groups (MEPS: SMD = −0.67; 95% CI, −1.79 to 0.44 and DASH: SMD = 0.37, 95% CI, −1.01 to 1.76).

#### 3.2.2. Range of Motions

Figure 3 shows the forest plots, SMD, 95% CI, and heterogeneity for the ROM. The estimated flexion-extension arc of the elbow was significantly higher in the TEA group than in the ORIF group (SMD = −0.45; 95% CI = −0.83 to −0.06; I^2^ = 15%). However, flexion (SMD = −0.35; 95% CI = −0.73, 0.03; I^2^ = 0%), loss of extension (SMD = 0.11; 95% CI = −0.27, 0.49; I^2^ = 33%), pronation (SMD = 0.35; 95% CI = −0.69, 1.39; I^2^ = 82%), and supination (SMD = −0.38; 95% CI = −1.01, 0.25; I^2^ = 55%) were not significantly different between the two groups.

#### 3.2.3. Complications and Reoperation

Seven studies [5,10,24,25,26,27,28] included in this meta-analysis reported the rate of total complications, such as wound dehiscence, heterotopic ossification, infection, elbow stiffness, and ulnar nerve problems. The data required for the analysis of the reoperation rate were also provided in all articles [5,10,20,21,24,25,26,27,28,29]. The results of our analysis suggested that the TEA group had a lower reoperation rate (pooled OR = 1.95; 95% CI = 1.26–3.00; I^2^ = 27%) than the ORIF group. However, there was no significant difference in the incidence of total complications (pooled OR = 1.65; 95% CI = 0.96–2.84; I^2^ = 0%) between the two groups (Figure 4).

Additionally, we performed an analysis of each complication. The incidence rate of elbow stiffness (pooled OR = 3.41; 95% CI = 1.09–10.68; I^2^ = 0%) after surgery was significantly lower in the TEA group than in the ORIF group. However, the incidence rates of wound dehiscence (pooled OR = 0.57; 95% CI = 0.16–2.09; I^2^ = 0%), heterotopic ossification (pooled OR = 0.47; 95% CI = 0.15–1.41; I^2^ = 8%), ulnar nerve problems (pooled OR = 2.07; 95% CI = 0.91–4.68; I^2^ = 20%), and infection (pooled OR = 0.78; 95% CI = 0.26–2.37; I^2^ = 0%) were not significantly different between the two groups (Figure 5).

#### 3.2.4. Publication Bias

A funnel plot analysis and Egger’s test were performed on the functional scores, range of motion, and complications. The *p*-value for all factors was >0.05. (MEPS, *p* = 0.1994; DASH, *p* = 0.3475; flexion/extension arc, *p* = 0.2631; flexion, *p* = 0.7231; extension, *p* = 0.9715; pronation, *p* = 0.5664; supination, *p* = 0.4154; reoperation, *p* = 0.1866; total complications, *p* = 0.5101; wound dehiscence, *p* = 0.4198; heterotopic ossification, *p* = 0.0518; ulnar nerve problems, *p* = 0.1025; infection, and *p* = 0.2103; and elbow stiffness, *p* = 0.1044).

## 4. Discussion

ORIF and TEA are the most popular surgical options for treating distal humeral fractures [5,12,20]. Only some clinical studies and a few meta-analyses have compared the functional outcomes and complications between the two methods [1,10,18,19,24,25,26,27]. However, previous meta-analyses have limitations owing to the lack of comparative studies. Several studies comparing these two methods have been published in the last three years [5,20,21]. Thus, an updated meta-analysis using recent comparative studies is necessary to overcome this problem. Therefore, we conducted the present meta-analysis, comparing ORIF and TEA for distal humeral fractures in elderly patients using 10 comparative studies. The flexion/extension arc showed that TEA produced significantly better outcomes than ORIF. In terms of complications, the rates of reoperation and elbow stiffness were significantly lower in the TEA group than in the ORIF group. However, the functional scores and rates of other complications were not significantly different between the two groups.

Achieving satisfactory functional outcomes after surgery is of great importance for the surgical management of distal humeral fractures. DASH and MEPS are the most frequently used scores for evaluating elbow function. A total of seven of the included studies [5,10,21,24,25,26,27] were evaluated using MEPS, and five studies used DASH; there were four studies, each with data available for analysis. Jordan et al. [1] reported that the MEPS and DASH scores for TEA patients were superior to those of ORIF patients. Two other meta-analyses [18,19] reported higher MEPS scores in the TEA and ORIF groups. In comparison, this meta-analysis showed no difference in the functional scores between the two groups.

The flexion/extension arc after surgery is a major concern for orthopedic surgeons. Most activities of daily living require from 30 to 130° of flexion [30]. In most of the included studies and meta-analyses, the TEA group showed a better mean flexion/extension arc [5,18,19,24,26]. The results of this analysis suggest that TEA produced significantly better outcomes in the flexion/extension arc than ORIF. In previous studies, statistically significant results could not be derived because of the small number of cases, but this analysis is thought to have overcome that issue.

In terms of complications, the flexion/extension arc is associated with elbow stiffness. Six of the included studies provided available data regarding the rate of elbow stiffness [31]. Baik et al. [5] defined elbow stiffness as a flexion of <120° and loss of extension of >30°. In other studies, the criteria for elbow stiffness were not described, and a large proportion of patients with elbow stiffness required a reoperation to resolve it. None of the previous meta-analyses evaluated the rate of elbow stiffness, and it was reported that none of the patients showed stiffness in the TEA among the included studies, except for one study.

In our meta-analysis, there was no significant difference in the total complication rate between the ORIF and TEA groups. A meta-analysis performed by Githens et al. [18] found a lower complication rate in the ORIF group than in the TEA group. (34.2% vs. 37.6%). The higher complication rate after TEA (than after ORIF) was also supported by Schindelar et al. [19] (25.0% vs. 17.0%). A recent comparison of the complication rates after ORIF and TEA has reported that there is no significant difference between the two groups; however, this study only evaluated the 30 days of the short-term [28].

The reoperation rates have been inconsistent in several of the studies. A systematic review reported by Schindelar et al. [28] found a higher reoperation rate for TEA than for ORIF (10.0% vs. 4.0%). Baik et al. [5] reported that the occurrence rate of major complications with a high probability of requiring reoperation was higher in the TEA group than in the ORIF group, which resulted in a relatively high reoperation rate in the TEA group. In comparison, Githens et al. [18] reported that reoperation rates were higher in the ORIF group than in the TEA group. The results of the analysis in this study suggest that the TEA group had a lower rate of reoperation than the ORIF group.

### Study Limitations

This study had several limitations. First, the quality of the studies included in this meta-analysis was not high. High-quality studies, such as prospective cohort studies and randomized controlled trials, are ideal for meta-analyses. The articles included in this study consisted of one randomized control study and nine retrospective comparative studies. Additionally, a relatively small number of cases were included in our analysis. For an accurate analysis, we included only 10 papers in which the number of experimental groups and control groups were clearly described; therefore, a relatively small number of papers were included. Common themes for study weaknesses included restricted information on the surgeons performing the surgery, the handling of missing data, perioperative care, the comorbid conditions of patients, and the details regarding patient selection. These factors are likely to have a major impact on the functional outcomes and complication rates.

## 5. Conclusions

In the current study, we compared two surgical options, ORIF and TEA, for the fixation of distal humeral fractures using a meta-analysis. TEA produced significantly better outcomes than ORIF with respect to the flexion/extension arc. In addition, the rates of reoperation and elbow stiffness were significantly lower in the TEA group than in the ORIF group. However, after surgery, other complication rates and functional scores (MEPS and DASH) were similar between the two groups. The outcomes from the present meta-analysis can provide guidance when selecting a surgical option for distal humeral fractures in the elderly.

## Figures and Tables

**Figure 1 jcm-11-05775-f001:**
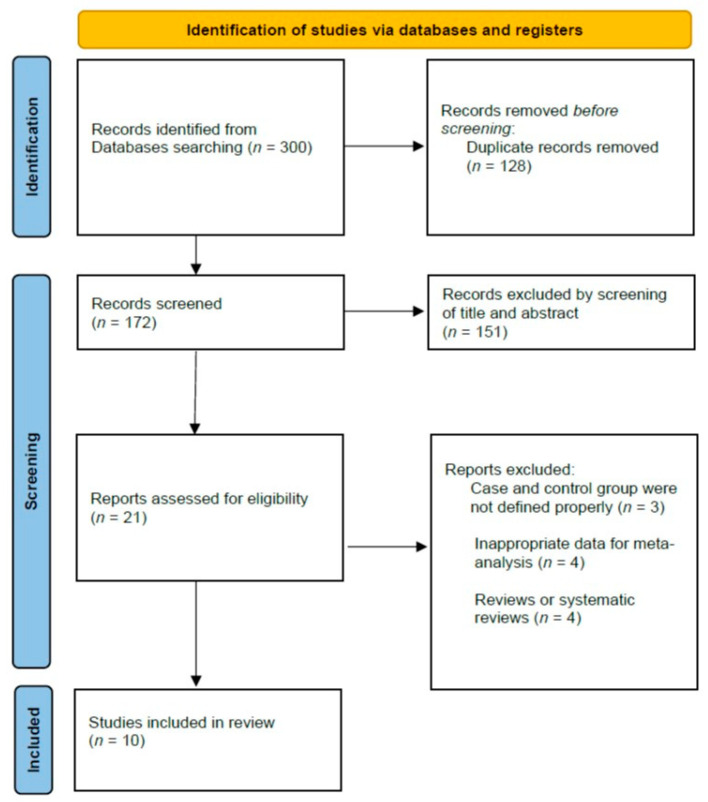
Flow chart for literature identification using the Preferred Reporting Items for Systematic Reviews and Meta-Analyses (PRISMA) guidelines.

**Figure 2 jcm-11-05775-f002:**
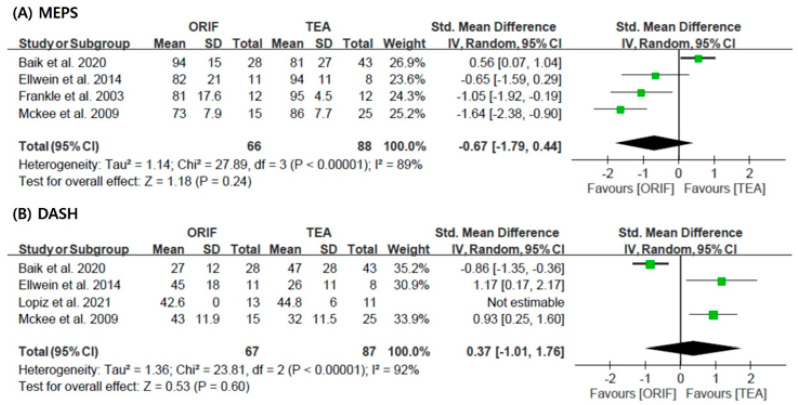
Results of the meta-analysis with respect to the functional scores: (**A**) MEPS, and (**B**) DASH [5,10,21,24,27]. MEPS: Mayo Elbow Performance score; DASH: Disabilities of the Arm, Shoulder, and Hand; ORIF: open reduction and internal fixation; TEA: total elbow arthroplasty; SD: standard deviation.

**Figure 3 jcm-11-05775-f003:**
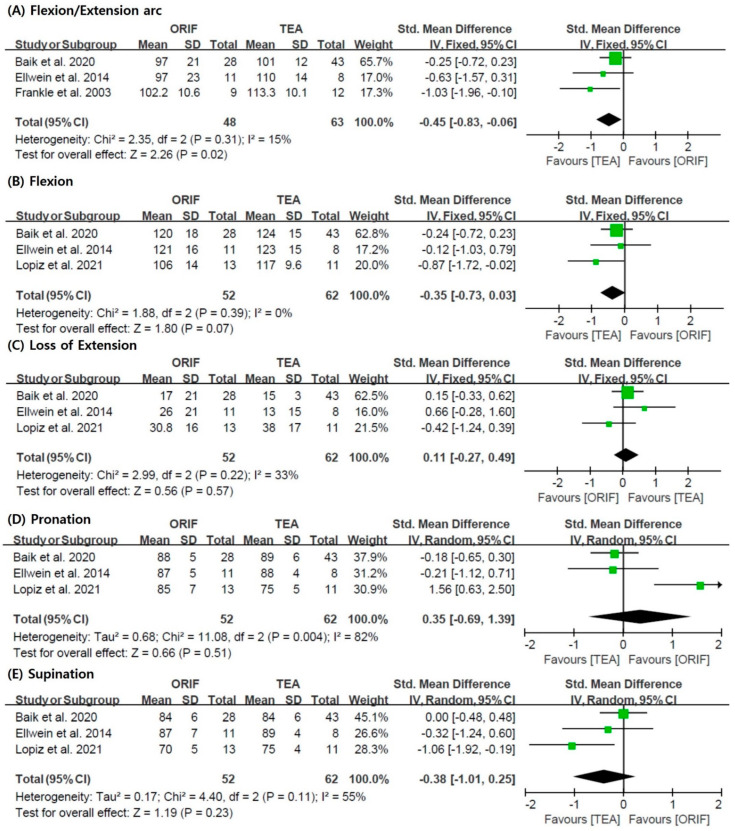
Results of the meta-analysis with respect to the range of motions: (**A**) flexion/extension arc, (**B**) flexion, (**C**) loss of extension, (**D**) pronation, and (**E**) supination [5,21,24,27]. ORIF: open reduction and internal fixation; TEA: total elbow arthroplasty; SD: standard deviation.

**Figure 4 jcm-11-05775-f004:**
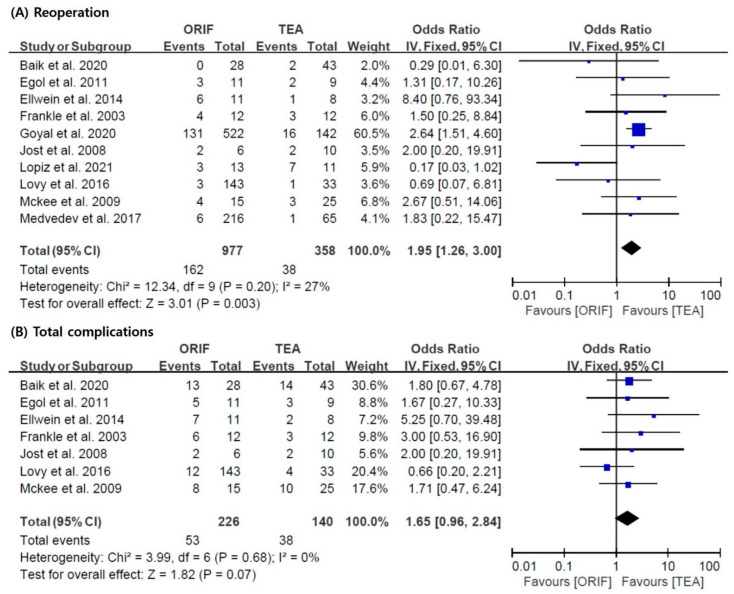
Results of the meta-analysis with respect to reoperation and total complications: (**A**) reoperation and (**B**) total complications [5,10,20,21,24,25,26,27,28,29]. ORIF: open reduction and internal fixation; TEA: total elbow arthroplasty.

**Figure 5 jcm-11-05775-f005:**
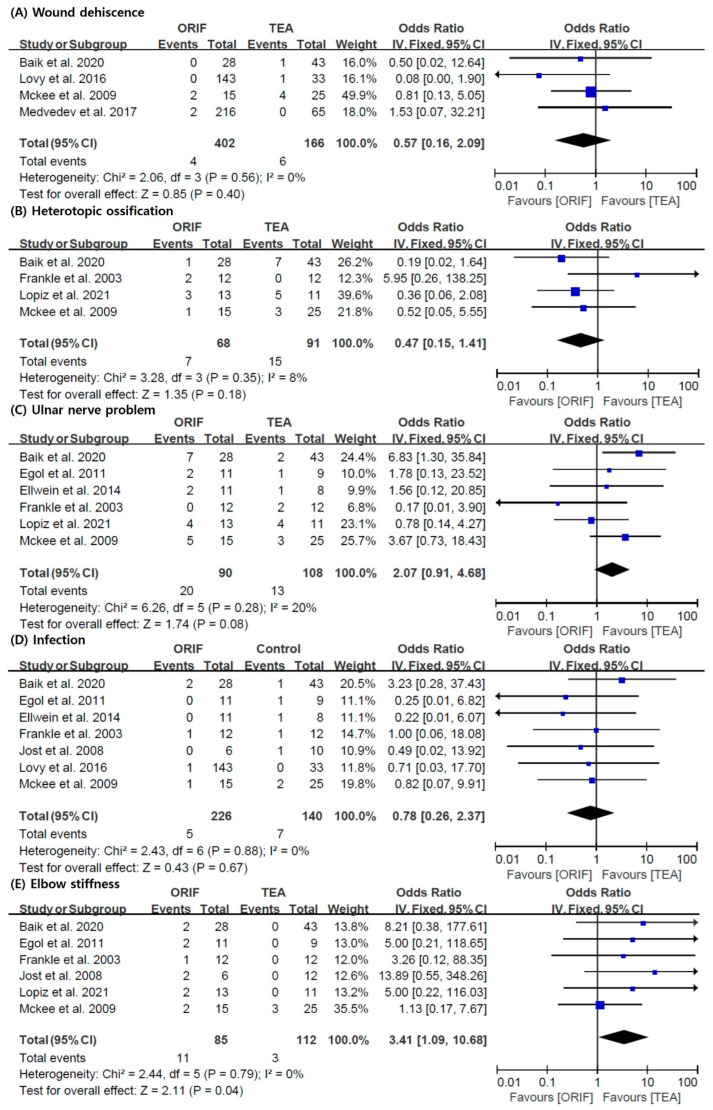
Results of the meta-analysis with respect to each complication: (**A**) wound dehiscence, (**B**) heterotopic ossification, (**C**) ulnar nerve problem, (**D**) infection, and (**E**) elbow stiffness [5,10,21,24,25,26,27,28,29]. ORIF: open reduction and internal fixation; TEA: total elbow arthroplasty.

**Table 1 jcm-11-05775-t001:** Characteristics of the included studies.

Authors (Year)	Study Design (LOE)	Mean Age, Years	Mean FU, mo	Fracture Type	ORIF, *n*	TEA, *n*	Total, *n*	Elbow Prosthesis	Device for ORIF	Outcomes Recorded	NOS
Frankle et al. (2003) [24]	Retrospective comparative (IV)	70.2	51	AO 13.C2, 13.C3	12	12	24	Coonrad-Morrey Semiconstrained (Zimmer)	Non-locking plate	MEPS, ROM, complications	8 (good)
Jost et al. (2008) [25]	Retrospective comparative (IV)	63.4	59.6	AO 13A, 13B, 13C	6	10	16	Coonrad-Morrey Semiconstrained (Zimmer)	Orthogonal or parallel configuration locking plates	MEPS, ROM, complications	7 (good)
Mckee et al. (2009) [10]	Randomized controlled trial (II)	77	24	AO 13C	15	25	40	Coonrad-Morrey Semiconstrained (Zimmer)	Orthogonal or parallel configuration locking plates	MEPS, DASH, complications	9 (good)
Egol et al. (2011) [26]	Retrospective comparative (IV)	77.4	14.8	AO 13B, 13C	11	9	20	Semiconstrained Implant (Coonrad-Morrey or Solar)	orthogonal or parallel configuration locking plates	MEPS, DASH, ROM, complications	7 (good)
Ellwein et al. (2014) [27]	Retrospective comparative (IV)	72	26	AO 13C	11	8	19	Semiconstrained, cemented (Latitude)	Orthogonal configuration locking plates	MEPS, DASH, ROM, complications	8 (good)
Lovy et al. (2016) [28]	Retrospective comparative (IV)	72.6	NR	ICD-9 812.2, 812.40, 812.41	143	33	176	NR	NR	complications	8 (good)
Medvedev et al. (2017) [29]	Retrospective comparative (IV)	78.1	NR	ICD-9 812.4x, 812.5x	216	65	281	NR	NR	complications	6 (good)
Baik et al. (2020) [5]	Retrospective comparative (IV)	77.8	32.8	AO 13C	28	43	71	Coonrad-Morrey semiconstrained prosthesis (Zimmer)	Double-locking plates	Pain, MEPS, DASH, ROM, complications	9 (good)
Goyal et al. (2020) [20]	Retrospective comparative (IV)	At least 65	NR	ICD-9: 812.40-3	522	142	664	NR	NR	complications	7 (good)
Lopiz et al. (2021) [21]	Retrospective comparative (IV)	80	64	AO 13C	13	11	24	Coonrad-Morrey semiconstrained (Zimmer) or the Link Endo-Model elbow prosthesis (Link^®^)	The Mayo Clinic Congruent elbow plate system (Acumed)	MEPS, DASH, ROM, complications	9 (good)

LOE: level of evidence; FU: follow-up; ORIF: open reduction and internal fixation; TEA: total elbow arthroplasty; NOS: Newcastle–Ottawa scale for meta-analysis; AO: Arbeitsgemeinschaft für osteosynthesefragen; ICD, International Classification of Diseases; MEPS, Mayo Elbow Performance score; DASH, Disabilities of the arm, shoulder, and hand; ROM: range of motion; NR: not recorded.

## Data Availability

Not applicable.

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
