# Peer review of "Comparison of the Complications, Reoperations, and Clinical Outcomes between Open Reduction and Internal Fixation and Total Elbow Arthroplasty for Distal Humeral Fractures in the Elderly: A Systematic Review and Meta-Analysis"

_jcm, 2022, doi:10.3390/jcm11195775_

Round 1
Reviewer 1 Report
The manuscript deals with a topic of great relevance in orthopaedic surgery. It is well written and authors had a meticulous attention to details. The methodological quality is also acceptable. However, revisions are needed:
-The introduction is clear and well-structured. However, authors should list indications, according to international guidelines, to the analyzed approaches;
-Materials and methods are complete and the authors make the study reproducible;
-Results are very detailed; however, authors should improve presentation to make understanding more fluent;
- Discussion ad a whole appears well written, if the authors deem it appropriate the could add doi: 10.4081/or.2020.8661 to line 201;
-Conclusions are in agreement with the results.
Author Response
The manuscript deals with a topic of great relevance in orthopaedic surgery. It is well written and authors had a meticulous attention to details. The methodological quality is also acceptable. However, revisions are needed:
-The introduction is clear and well-structured. However, authors should list indications, according to international guidelines, to the analyzed approaches;
A) we revised the introduction part as you commented
(Line 38-43)
-Materials and methods are complete and the authors make the study reproducible;
A) N/A
-Results are very detailed; however, authors should improve presentation to make understanding more fluent;
A) correction of the manuscript was performed by professional once at the time of initial submission, but it was performed once more for details.
- Discussion ad a whole appears well written, if the authors deem it appropriate the could add doi: 10.4081/or.2020.8661 to line 201;
A) Thanks for the nice comment. Line 206 was added as the mentioned paper was judged to be suitable.(Reference 31)
-Conclusions are in agreement with the results.
A) N/A

Reviewer 2 Report
Thank you for letting me review this interesting systematic review of the best treatment of distal humeral fractures in the elderly.
The manuscript is well-written, especially the Conclusion-subhead at the end and the bullet points in terms of the most important findings. This study makes it possible to clearly validate one type of technique rather than another with sufficient difference in the result.
The authors state that PRISMA is followed, which in my opinion is absolutely necessary. Unfortunately they did not report whether they registered their meta-analysis, for example in PROSPERO.
Abstract
Line 14: Good that you used all these databases!
Line 19: The sentence on the p-value could be earsed in the abstract, this is common usage.
Line 25: Maybe “Elderly” should be used as keyword also.
Introduction:
In my opinion the Introduction is rather short. I would also appreciate a better presentation of the meta-analysis done on that subject before.
Methods:
Methodology is clear and precise.
Line 50: Could you please provide the exact search strings that you used – maybe as an additional table?
Line 55-59: Did you screen reference lists from included studies and reviews?
Line 73ff: Would it be possible to rate the quality of the included studies according to MINORS – in my opinion this methodological score is much better evaluated.
Results:
Table 1: Would it be possible to translate the ICD-9 into a fracture type according to AO? Please provide the level of evidence for the included studies.
Figure 2: There is a typo in “functional scores”
Discussion:
Too long, especially compared with the introduction.
Line 185-207: Why do you think that the TEA provided a better F/E arc but that did not influence the functional scores? Give reasons.
Please try to reason why your results differ from that of older meta-analysis. Is it just the different study design that you included?
Please include a comment on the different methods of ORIF (locking vs. non-locking). In my opinion that plays an important role in the performance of ORIF and 4 out of 10 studies used non-locking or did not report that. Maybe you could get that information from the authors of those NR-studies?
Author Response
Please see the attachment
Thank you for letting me review this interesting systematic review of the best treatment of distal humeral fractures in the elderly.
The manuscript is well-written, especially the Conclusion-subhead at the end and the bullet points in terms of the most important findings. This study makes it possible to clearly validate one type of technique rather than another with sufficient difference in the result.
The authors state that PRISMA is followed, which in my opinion is absolutely necessary. Unfortunately they did not report whether they registered their meta-analysis, for example in PROSPERO.
Abstract
Line 14: Good that you used all these databases!
Line 19: The sentence on the p-value could be earsed in the abstract, this is common usage.
- We erased it
Line 25: Maybe “Elderly” should be used as keyword also.
- We added it to keywords (Line 25)
Introduction:
In my opinion the Introduction is rather short. I would also appreciate a better presentation of the meta-analysis done on that subject before.
- We revise introduction part as you commented (Line 38-43)
Methods:
Methodology is clear and precise.
Line 50: Could you please provide the exact search strings that you used – maybe as an additional table?\
- The term mentioned in lines 57-59 is the search term in the actual search, the process is detailed in figure
Line 55-59: Did you screen reference lists from included studies and reviews?
- We did.
Line 73: Would it be possible to rate the quality of the included studies according to MINORS – in my opinion this methodological score is much better evaluated.
- The Newcastle–Ottawa Scale is widely used to evaluate studies included in meta-analysis, so it is expected that there will be no problem.
Results:
Table 1: Would it be possible to translate the ICD-9 into a fracture type according to AO? Please provide the level of evidence for the included studies.
- It is difficult to accurately match ICD-9 and AO classification. We added level of evidence for included studies.
Figure 2: There is a typo in “functional scores”
- A) Correction is done (functionasl -> functional) : Line 142
Discussion:
Too long, especially compared with the introduction.
Line 185-207: Why do you think that the TEA provided a better F/E arc but that did not influence the functional scores? Give reasons.
- A) As mentioned in the limitation, it is difficult to fully consider factors such as cormorbidity that affect the functional score, so this result is thought to have occurred.
Please try to reason why your results differ from that of older meta-analysis. Is it just the different study design that you included?
- A) As mentioned in the manuscript, good quality meta-analysis has not been performed before due to the lack of comparative studies. Therefore, this study was planned, and this is considered to be the decisive reason to have different results from previous studies.
Please include a comment on the different methods of ORIF (locking vs. non-locking). In my opinion that plays an important role in the performance of ORIF and 4 out of 10 studies used non-locking or did not report that. Maybe you could get that information from the authors of those NR-studies?
- A) I think that's a good idea. The difference between locking-non locking is also judged to have an effect on the postoperative outcome, which the authors considered. Whether to exclude NR-study was also considered, but it was judged that it would not have a significant impact on other analyses. Some of the authors of included studies contacted for information but did not receive a reply.

Reviewer 3 Report
The subject is interesting and in my opinion clinically relevant.
The abstract do not mention the absence of difference in terms of functional score : this should appears clearly in the abstract. The same way, the benefits in flexion/extension are clearly mentioned in the abstract but there is no difference in the results concerning : flexion, loss of extension, pronation and supination. The statement in the abstract doesn't reflect the reality of the paper. this should be changed.
The Material and methods part is perfect
In the results parts, the figures are not easy to understand for the reader. Probably better to change the figure type to make it easier to understand. Mentioning the range of motion in degrees in the text would be interesting
Discussion : the limitation part is clear.
Conclusion :
Congratulation for this interesting work.
Author Response
Please see the attachment
The subject is interesting and in my opinion clinically relevant.
---The abstract do not mention the absence of difference in terms of functional score : this should appears clearly in the abstract. The same way, the benefits in flexion/extension are clearly mentioned in the abstract but there is no difference in the results concerning : flexion, loss of extension, pronation and supination. The statement in the abstract doesn't reflect the reality of the paper. this should be changed.
- Thank you for nice comments and sincere feedback. I changed the abstract as you commented. Added sentences about functional scores and other ROMs such as other flexions, extensions, and pronation. (line 21-23)
---The Material and methods part is perfect
----In the results parts, the figures are not easy to understand for the reader. Probably better to change the figure type to make it easier to understand. Mentioning the range of motion in degrees in the text would be interesting
- That's a good point. However, in terms of figures, it is judged that there will be no difficulty in understanding the figure, as it is the format most used in the meta-analysis, with little effort for authors who can read this paper. Likewise, the mention of the exact angle in the range of motion is also not considered to be very important in meta-analysis.
---Discussion : the limitation part is clear.
---Conclusion :
---Congratulation for this interesting work.
